# Longitudinal Study Comparing Mental Health Outcomes in Frontline Emergency Department Healthcare Workers through the Different Waves of the COVID-19 Pandemic

**DOI:** 10.3390/ijerph192416878

**Published:** 2022-12-15

**Authors:** Francesca Th’ng, Kailing Adriel Rao, Lixia Ge, Hwee Nah Neo, Joseph Antonio De Molina, Wei Yang Lim, Desmond Mao

**Affiliations:** 1Acute & Emergency Care Department, Khoo Teck Puat Hospital, 90 Yishun Central, Singapore 768828, Singapore; 2Health Services and Outcomes Research, National Healthcare Group, 3 Fusionopolis Link, Singapore 138543, Singapore; 3School of Medicine, University of Galway, University Road, Galway H91TK33, Ireland

**Keywords:** COVID-19, depression, anxiety, stress, post-traumatic stress disorder, healthcare workers, emergency department, mental health

## Abstract

As countries transition from the COVID-19 pandemic to endemic status, healthcare systems continue to be under pressure. We aimed to quantify changes in depression, anxiety, stress and post-traumatic stress disorder (PTSD) between 3 cohorts (2020, 2021 and 2022) of our Emergency Department (ED) healthcare workers (HCWs) and those who had worked through all 3 phases of the pandemic; and identify factors associated with poorer mental health outcomes (MHOs). In this longitudinal single-centre study in Singapore, three surveys were carried out yearly (2020, 2021 and 2022) since the COVID-19 outbreak. Depression, anxiety and stress were measured using DASS-21, and PTSD was measured using IES-R. A total of 327 HCWs (90.1%) participated in 2020, 279 (71.5%) in 2021 and 397 (92.8%) in 2022. In 2022, ED HCWs had greater concerns about workload (Mean score ± SD: 2022: 4.81 ± 0.86, vs. 2021: 4.37 ± 0.89, vs. 2020: 4.04 ± 0.97) and perceived to have less workplace support (2022: 4.48 ± 0.76, vs. 2021: 4.66 ± 0.70, vs. 2020: 4.80 ± 0.69). There was overall worsening depression (27.5% in 2020, 29.7% in 2021 and 32.2% in 2022) and stress (12.2% in 2020, 14.0% in 2021 and 17.4% in 2022). Healthcare assistants as a subgroup had improving MHOs. ED HCWs who were female and had psychiatric history, were living with the elderly, and had concerns about their working environment, workload and infection had poorer MHOs. This study will guide us in refining existing and devising more focused interventions to further support our ED HCWs’ wellbeing.

## 1. Introduction

After about a 3.5-year long battle with the COVID-19 outbreak, many countries are taking a new approach: learning to live with COVID-19 rather than eradicating it. Similarly, Singapore is moving towards endemicity and is scaling back on infection control restrictions [1]. As we transition into the endemic phase of COVID-19, our healthcare system continues to be under pressure, and the wellbeing of our healthcare workers (HCWs) is more important than ever before [2].

Currently, there are limited longitudinal studies monitoring mental health changes in HCWs as we transition from the COVID-19 pandemic to an endemic state. A study amongst six Southeast Asian countries (Indonesia, Malaysia, Philippines, Singapore, Thailand and Vietnam) in 2021 found that job burnout rates were highest across the countries, followed by anxiety and depression [3]. Anxiety was found to be higher (10%) than pre-COVID-19 (2.2–4.9%). In their study, longer-than-usual working hours, the perception of high risk from COVID-19 infection and the inadequacy of personal protective equipment (PPE) were associated with higher odds of burnout and anxiety. Protective factors like good teamwork were associated with lower odds of burnout, anxiety and depression.

We had previously conducted two studies amongst our Emergency Department (ED) HCWs in 2020 [4] and 2021 [5]. The 2021 results showed improvement in anxiety amongst ED HCWs and worsening depression amongst ED doctors over one year. Age, living with the elderly and concerns about workload and infection risk were associated with higher odds of depression and anxiety. Following our second survey in 2021, efforts have been made by our ED and the hospital to improve HCWs’ wellbeing. These included the creation of a departmental wellbeing committee to provide support for our HCWs; HCWs with families overseas being granted leave to see their loved ones; and the hospital’s initiation of Staff Wellness Passes for extra protected time off from work. As Singapore transitioned towards an endemic status at the beginning of 2022, there have been significant national changes in COVID-19 safety measures, including no restrictions on dining out and the opening up of travel borders (Table A1).

Leveraging on the prior studies conducted, our 3rd survey aimed to (1) quantify changes in MHOs (depression, anxiety, stress and post-traumatic stress disorder (PTSD) symptoms) between 3 cohorts of our frontline ED HCWs in 2020, 2021 and 2022; (2) quantify changes in MHOs between our ED HCWs who were working through all 3 phases of the pandemic; and (3) identify the factors longitudinally associated with poorer MHOs. We hypothesised that MHOs would generally improve amongst our ED HCWs, as they would have adapted to the changes that have occurred over the past 3.5 years and as the nation eased infection control measures.

## 2. Materials and Methods

### 2.1. Study Design and Participants

This is a prospective longitudinal study carried out amongst frontline ED HCWs in Khoo Teck Puat Hospital, Singapore. The study hospital is a 795-bed acute hospital that serves more than 550,000 people living in the north of Singapore. The average number of ED patients seen, including the number of P1 (triage acuity level 1) and P2 (triage acuity level 2) patients per month, the average waiting time for ED patients to be seen by a doctor and the average waiting time for admitted patients to obtain a ward bed in 2019 to 2022 are illustrated in Table A2.

Three rounds of surveys had been carried out annually since the outbreak of COVID-19 in Singapore in 2020. The first survey was conducted in June 2020 [4] during the first wave of the pandemic; the second one was conducted one year later in June 2021 [5] when there was an emergence of COVID-19 variants, including Omicron; and the third survey was carried out in June 2022 as Singapore transitioned from a COVID-19 pandemic to endemic status. The methodologies for the first two surveys have been described previously [4,5]. In this third survey, all ED HCWs of KTPH were invited to complete a paper-based survey questionnaire (Figure A1). Participation was voluntary, and written consent was obtained. Participants returned the completed questionnaires to the investigators either at the end of their work shifts in person or dropped them off directly into a collection box at the ED office. The three surveys were reviewed and approved independently by the National Healthcare Group Domain Specific Review Board (DSRB 2020/00653, 2021/00336, and 2022/00290). Data of participants who participated in the three surveys were matched based on their reference numbers (last four digits of handphone number) with reference to the demographics (e.g., age, gender and ethnicity) and occupation to ensure accuracy.

### 2.2. Outcome Measures

Data for depression, anxiety and stress were collected using 21-item validated Depression, Anxiety and Stress Scale (DASS-21) [6], similar to our previous 2 studies. The sum score for each MHO was calculated and multiplied by two, which was then used to categorise individual MHO into two groups (normal vs. positive for each MHO). A positive score for depression, anxiety and stress was defined as >9, >7 and >14, respectively. PTSD was measured using 22-item Impact of Events Scale-Revised (IES-R) scale [7]. A cut-off score of ≥24 was used to define PTSD symptoms of clinical concern.

All HCWs’ demographic information, including age group, gender, ethnicity, occupation and living arrangement were collected in all three surveys. HCWs’ concerns and perceptions were collected using a questionnaire containing a list of statements based on a Likert scale where 1 = Strongly Disagree and 6 = Strongly Agree. This questionnaire was developed based on experts’ opinions (study team’s ED consultants, senior nurse and biostatisticians). These concerns and perception statements were categorised into five domains based on their content relevance and inter-item correlations (Figure A2), namely concerns about COVID-19 infection risk, perceptions about workplace support, concerns about workload, concerns about working environment and perception about how socially connected they were. The responses for perception about religion and exercise as a way of coping with stress and whether they felt respected were re-categorised into binary variables: “Yes” for responses of “Not sure but probably agree”, “Agree”, and “Strongly agree”, and “No” for responses of “Not sure but probably disagree”, “Disagree”, and “Strongly disagree”. The word ‘trend’ is used in the manuscript to describe the direction of change a variable is moving towards and does not equate to statistical significance.

### 2.3. Statistical Analysis

We performed analysis on three cohorts as well as on matched HCWs. The three cohorts referred to the HCWs who had responded to the respective year’s survey. The matched HCWs referred to the 160 HCWs who had responded to all 3 surveys. Categorical variables were described using frequencies and percentages, and continuous variables were expressed in means and standard deviations (SD), and medians and 1st and 3rd quartiles (Q1-Q3). The distribution of the severity and status of each MHO, as well as MHOs and concerns and perception scores for the three cohorts, were visually compared without using statistical tests, as the data were partially dependent. For the matched HCWs, Repeated Measure ANOVA was conducted to compare scores of concerns and perception since they were normally distributed. An unadjusted Generalized Estimating Equation (GEE) [8,9] by specifying binominal family and logit link function was performed for each MHO status to test whether survey year was associated with any MHO status. GEE approach facilitates analysis of longitudinal data or repeated measures designs and produces more efficient and unbiased regression estimates, as it takes into account the correlation of within-subject data.

GEE employing binominal family and logit link function was also performed to identify potential factors that were associated with risk of individual MHO for the three cohorts. In each model, status of one MHO (binary variable) was the dependent variable. The survey year, characteristics, domain scores of concerns about COVID-19 infection, working environment and workload, perceptions about workplace support and social connectedness, two coping items (binary) and feeling respected (binary), which were identified to be associated with any MHO (*p* < 0.1) in univariate GEE analyses, were included as independent variables in the final model.

Odds ratios (OR) when outcome variable is binary or beta coefficients when outcome variable is continuous 95% confidence intervals (CIs) and *p*-values were reported. All analyses were performed using Stata/SE 16.1. *p* < 0.05 was set as the level of significance.

## 3. Results

### 3.1. Demographics of the Cohorts

The response rate for each round of the survey was: 90.1% in 2020, 71.5% in 2021 and 92.8% in 2022. A total of 160 ED HCWs participated in all three rounds (Figure 1). Table 1 shows the demographic characteristics of all ED HCWs in 2020, 2021 and 2022, respectively, and Table A3 shows the demographic characteristics of the 160 matched HCWs. In comparison to 2020 and 2021, the cohort in 2022 was generally younger and had a higher proportion of HCAs, a lower proportion of married HCWs and a higher proportion of HCWs with a family member(s) or friend(s) who had contracted COVID-19 (Table 1).

### 3.2. Concerns and Perceptions of the Cohorts

Overall, ED HCWs in 2022 had greater concerns about their workload (mean score ± SD: 2022: 4.81 ± 0.86, vs. 2021: 4.37 ± 0.89, vs. 2020: 4.04 ± 0.97) and perceived to have less workplace support (2022: 4.48 ± 0.76, vs. 2021: 4.66 ± 0.70, vs. 2020: 4.80 ± 0.69) (Table 2). In 2022, the ED HCWs had fewer concerns about COVID-19 infection risk (2022: 3.90 ± 0.92, vs. 2021: 3.93 ± 0.83, vs. 2020: 4.19 ± 0.82) and their working environment (2022: 3.90 ± 1.03, vs. 2021: 3.96 ± 0.98, vs. 2020: 4.09 ± 0.85). Similarly, the 160 matched ED HCWs followed parallel trends in these categories (Table A4).

### 3.3. Mental Health Outcomes

#### 3.3.1. Depression

A total of 27.5% of ED HCWs screened positive for depression in 2020, 29.7% in 2021 and 32.2% in 2022 (Table 3), reflecting an increasing trend. The score distribution for each MHO in each cohort is shown in Figure A3. In the HCA group, however, the trend was reversed (2020: 52.4%, vs. 2021: 33.3%, vs. 2022: 30.2%). The score distribution for each MHO in each cohort is shown in Figure A3. The unadjusted GEE with each MHO score as the outcome showed that amongst the matched HCWs, the proportion who screened positive for depression (Table A5) and their depression scores (Table A5) remained similar.

The GEE results (Table 4) showed that junior doctors (OR [95%CI]: 0.43 [0.19,0.99], *p* = 0.048), those with a greater number of years in their occupation (OR 0.94 [0.90,0.98], *p* = 0.005), those who perceived that they had better workplace support (OR 0.74 [0.57,0.96], *p* = 0.021), those who were socially connected (OR 0.50 [0.38,0.64], *p* < 0.001) and those who felt respected (OR 0.63 [0.42,0.95], *p* = 0.026) had lower odds of developing depression (Table 4). ED HCWs with a psychiatric history (OR 3.75 [1.41,9.96], *p* = 0.008], those who were living with the elderly (OR 1.82 [1.20,2.77], *p* = 0.005) and those with concerns about their working environment (OR 1.21 [1.03,1.44], *p* = 0.024) and workload (OR 1.46 [1.19,1.79], *p* < 0.001) had higher odds of developing depression. Compared to the 2020 cohort, the odds of developing depression in the 2021 and 2022 cohorts were lower but not significant.

#### 3.3.2. Anxiety

A total of 34.2% of ED HCWs screened positive for anxiety in 2020, 28.7% in 2021 and 38.5% in 2022 (Table 3). There was a reduction in the proportion of HCAs who screened positive for anxiety (2020: 71.4%, vs. 2021: 66.7%, vs. 2022: 49.1%) and a reduction in the HCAs’ anxiety scores (2020: 6.0 ± 4.1, vs. 2021: 4.6 ± 2.3, vs. 2022: 3.9 ± 3.5) (Table 3). Improvement in anxiety was also observed amongst the matched HCAs (2020: 71.4%, vs. 2021: 57.1%, vs. 2022: 42.9%) (Table A5). Amongst the 121 nursing staff who participated in all three surveys, there was a reduction in the risk of developing anxiety in 2021 and 2022 compared to 2020 (Table A6).

The GEE results showed that ED HCWs who were 31–40 years old (OR 0.66 [0.45,0.97], *p* = 0.036), who perceived themselves to be socially connected (OR 0.67 [0.52,0.85], *p* = 0.001) and felt respected (OR 0.56 [0.38,0.84], *p* = 0.005) had lower odds of developing anxiety (Table 4). ED HCWs who were female (OR 1.86 [1.20,2.89], *p* = 0.005), with concerns about infection risk (OR 1.31 [1.06,1.62], *p* = 0.011) and working environment (OR 1.28 [1.08,1.51], *p* = 0.004) had higher odds of developing anxiety. Compared to the 2020 cohort, the odds of developing anxiety in the 2021 and 2022 cohorts were lower but not significant.

#### 3.3.3. Stress

A total of 12.2% of ED HCWs screened positive for stress in 2020, which increased to 14% in 2021 and 17.4% in 2022 (Table 3). The proportion of junior doctors (2020: 4.7%, vs. 2021: 13.2%, vs. 2022: 15.7%) and nurses (2020: 12.0%, vs. 2021: 14.6%, vs. 2022: 19.7%) who screened positive for stress was increasing. The stress scores amongst senior doctors, junior doctors and nurses were also increasing. In contrast, there was a reduction in stress scores amongst HCAs (Mean ± SD: 2020: 6.7 ± 4.0, vs. 2021: 4.8 ± 1.9, vs. 2022: 4.4 ± 3.9) (Table 3).

GEE results showed that ED HCWs with better social connections (OR 0.57 [0.44,0.75], *p* < 0.001) and who felt respected (OR 0.56 [0.34,0.93], *p* = 0.026) had lower odds of developing stress (Table 4). ED HCWs who had a psychiatric history (OR 3.20 [1.15,8.92], *p* = 0.027), those who were living with elderly (OR 1.71 [1.08,2.70], *p* = 0.022) and had concerns about workload (OR 1.90 [1.41,2.55], *p* < 0.001) had higher odds of developing stress. Compared to the 2020 cohort, the odds of developing stress in the 2021 and 2022 cohorts were lower but not significant.

#### 3.3.4. PTSD of Clinical Concern

A total of 16.2% of ED HCWs screened positive for PTSD in 2020, 13.6% in 2021 and 16.1% in 2022 (Table 3). There was a downward trend in PTSD scores among HCAs (mean ± SD: 2020: 25.1 ± 17.5, vs. 2021: 17.3 ± 9.7, vs. 2022: 15.5 ± 13.1) (Table 3).

The GEE results showed that ED HCWs who perceived themselves to have better social connections (OR 0.52 [0.39,0.70], *p* < 0.001) had lower odds of developing PSTD (Table 4). ED HCWs who were living with the elderly (OR 2.12 [1.32,3.40], *p* = 0.002) and had concerns about infection risk (OR 1.48 [1.15,1.92], *p* = 0.003) and workload (OR 1.64 [1.25,2.15], *p* < 0.001) had higher odds of developing PTSD. Compared to the 2020 cohort, the odds of developing PTSD of clinical concern in the 2021 and 2022 cohorts were lower but not significant.

## 4. Discussion

Our 3-year prospective cohort study found (1) worsening depression and stress in the overall cohort, (2) improving anxiety, stress and PTSD scores amongst HCAs as a subgroup, (3) increased concerns about workload, (4) an overall perception of receiving less workplace support and (5) reduced concerns about COVID-19 infection risk and working environment. ED HCWs who were female, had a psychiatric history, were living with the elderly and had concerns about the working environment, workload and infection risk had poorer MHOs.

### 4.1. Overall Worsening Depression and Stress

Overall, there was an increasing proportion of ED HCWs who screened positive for depression and stress, and their scores were increasing over the years; these were not statistically significant when adjusted for (Table 4). Nevertheless, these are interesting findings, as we had expected ED HCWs to have received care for their mental health concerns or to have psychologically adapted to the changes within the healthcare system and community over the past 3.5 years. These were also in spite of the easing of infection-control measures nationally since the beginning of 2022 and efforts by the hospital and department to improve HCWs’ wellbeing. The prevalence of depression (27.5–32.3%) amongst our cohorts of ED HCWs is much higher than that demonstrated by Teo et. al.’s study [3], which was carried out across 6 Southeast Asian countries (an average of 4%). This could be due to the cohort sampling differences, as Teo et. al.’s study included other non-frontline HCWs - EMTs and hospital administrative staff, and had used different measurement tools for depression. Nevertheless, their study showed that Singapore HCWs reported the highest levels of burnout (39%), anxiety (21%) and depression (9%) compared to the 5 other countries.

In line with our study’s findings, a cross-sectional study [10] amongst Taiwanese frontline HCWs showed persistently poor MHOs (anxiety, depression and insomnia) irrespective of the wave of the pandemic. This was partly explained by the changes in workload, work schedules, working overtime and concerns over the risk of infection [10,11]. Similarly, a Chinese study that was carried out about 3 years after the 2003 severe acute respiratory syndrome (SARS) outbreak showed persistently high levels of psychological stress, which was thought to be attributed to working in a high-risk environment and having a fear of being a source of infection to a family member(s) [12]. These factors, specifically concerns about workload and the working environment, could likely explain the persistently poor levels of depression and stress amongst our ED HCWs.

However, our subgroup of HCAs bucked this trend and had improved MHOs over the years; there was a reduction in the proportion of HCAs with depression, anxiety and PTSD and an improvement in anxiety, stress and PTSD scores. From 2020 to 2022, there was a considerable increase in the number of HCAs recruited by the department (Table 1). We postulate several reasons for HCAs’ improved MHOs: Firstly, new HCAs voluntarily joined the department during the pandemic and hence would have likely been adequately self-educated on COVID-19 and psychologically prepared for the type of work they would carry out and the working environment they would be in. Secondly, with the increase in workforce numbers, patient care and workload could be distributed appropriately and thus easing the burden off each other. They would also be able to provide more camaraderie and social support to one another, helping to alleviate uncertainties and their concerns about infection risk and the working environment. The job description of HCAs includes taking patients’ vital signs, doing point-of-care tests and tending to patients’ hygiene and personal care. The work is generally less intense compared to nurses.

### 4.2. Concerns about Workload and Workplace Support

Workload-related concerns have grown from 2020 to 2022 across our subgroups of HCWs. The pandemic has placed great pressure on the healthcare system, and many studies have attributed that to a combination of an increase in workload and the attrition of HCWs [13]. HCWs suffer from stress and burnout when overworked, and that compromises their ability to deliver good care [14]. Despite being in an endemic phase, there will be intermittent surges in COVID-19 patients attending healthcare services with the ongoing emergence of different COVID-19 variants and the resuming of normal social activities [15,16]. Interestingly, in spite of the lower ED attendances when compared to pre-COVID-19 numbers, there had been an increasing number of sicker patients requiring higher acuity care (Table A2). Waiting time to see a doctor stayed fairly constant, yet the average waiting time to obtain a ward bed has increased drastically. This access block issue has caused the ED to be overcrowded, and the issue is also evident in other public hospitals nationally [17,18]. ED HCWs do not just have to tend to new incoming patients but also patients who are boarding in the ED. Furthermore, the COVID-19 pandemic has affected the processes of routine comprehensive care for chronic patients due to the repurposing of healthcare facilities and reduction in services [19]. This in turn has resulted in a possible “rebound effect” of non-COVID-19 patients presenting to the ED. Judging from the increase in higher-acuity patients received in our ED, we should perhaps give some thought to the impact on the healthcare system as we deal with the aftermath of suboptimally managed chronic diseases after the pandemic phase is over [20]. Overall, even though the staff numbers in our department have grown these 3 years, which was largely caused by the increased hire of HCAs, the more experienced workforce at the start of the pandemic was replaced by new hires (evidenced by changes in staff demographics in Table 1), further contributing to the persistent high scores on workload concerns.

There was also the perception of less support from supervisors and colleagues in spite of measures being put in place by the hospital to provide mental support to HCWs in the form of wellness programs and the provision of a care hotline. We believe this perception has much to do with the factors outlined above, and it goes beyond just increasing the healthcare workforce numbers. The new hires may consist of redeployed staff, who would have to match the skillsets of what needs to be done. Recent publications on staff redeployment during the pandemic have highlighted the importance of carrying out detailed skills assessment to ensure patients’ needs are met [21]. In addition, access blocks and ED overcrowding erode staff resilience and contribute to staff feeling unsafe and unsupported [22]. Solving such operational issues will likely have a greater impact on staff wellbeing than just the provision of wellness programs.

### 4.3. Whole Sampled Cohort vs. Matched Cohort

When we delved into the differences between the 2 groups, we realised that there were slight differences. MHOs in the 160 matched cohort showed similar trends from 2020 to 2022 in both severity and scores (Table A5). This was in comparison to the overall worsening of depression and stress when we looked at the entire sampled population. Another interesting finding in the matched cohort (Table A6) is that the odds of having anxiety in 2021 and 2022 were lower compared to 2020 (2021: OR 0.67 [0.46–0.99] vs. 2022: OR 0.63 [0.42–0.93]) and is most evident amongst the nursing staff. We were not surprised that this finding occurred in the matched cohort, who had been through all 3.5 years of the pandemic. Anxiety in this group would have improved from 2020 when information and knowledge of the pandemic became more available through the subsequent years.

### 4.4. Strengths and Limitations

To our knowledge, this is one of the few studies to assess MHOs amongst ED HCWs over different waves of COVID-19. Validated assessment tools were used to measure MHOs. Our study analysed the ED cohort as a whole and those who had completed 3 surveys (matched). Most similar longitudinal studies just compared cohorts from the same place of interest [23,24]. With this information, we can target more focused interventions and prevention measures for the HCWs who have been with the department for the last 3.5 years, as well as new hires.

The limitations of this study include it being a single-centre study, which may limit the study’s generalisability to other healthcare settings. Voluntary participation and the lower response rate in 2021 could potentially have introduced selection bias. Only known confounders were corrected for. Socioeconomic factors, for example, housing conditions, which could have been confounders, were not included. The self-reporting nature of DASS-21 and IES-R, rather than clinician-facilitated assessments, could have also introduced bias.

## 5. Conclusions

In summary, our study showed that our frontline ED HCWs continue to have overall poor levels of depression, anxiety, stress and PTSD, irrespective of the wave of the pandemic. There was worsening depression and stress in the entire cohort, with the exception of the HCAs, for the various reasons mentioned above. ED HCWs who were female, had a psychiatric history, who were living with the elderly and had concerns about the working environment, workload and infection risk had poorer MHOs.

This study is crucial in aiding healthcare systems to identify potentially modifiable workplace factors associated with poorer MHOs. These will guide us in refining existing and in devising more focused interventions to further support our ED HCWs’ wellbeing. Furthermore, the insights gleaned from this study about HCWs’ concerns about workload and workplace support will aid us in optimising workflow processes with regards to the access block problems of staff attrition and staff redeployment in order to build a more resilient frontline workforce. It will be interesting and beneficial to our ED HCWs and to the wider national healthcare system to further reassess the changes in their MHOs over the next few years as the pandemic settles into endemicity.

## Figures and Tables

**Figure 1 ijerph-19-16878-f001:**
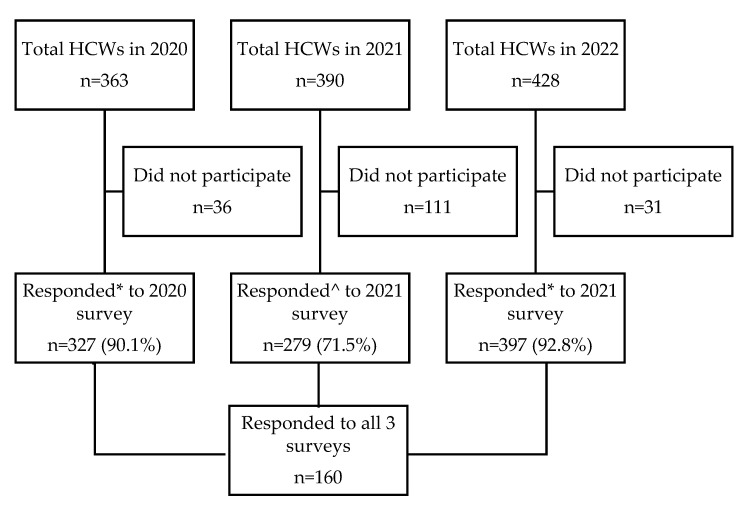
Flow diagram of the COVID-19 studies carried out in June 2020 (1st survey), June 2021 (2nd survey) and June 2022 (3rd survey). * All ED HCWs at the time of the study’s recruitment period were approached for recruitment. ^ Only ED HCWs who had participated in the first survey in 2020 were approached.

**Table 1 ijerph-19-16878-t001:** Characteristics of all ED HCWs in the 2020 (n = 327), 2021 (n = 279) and 2022 (n = 397) cohorts.

Characteristics	2020 (n = 327)	2021 (n = 279)	2022 (n = 397)
**Age group in years (n,%)**			
21–30	154 (47.1)	110 (39.4)	210 (52.9)
31–40	121 (37.0)	124 (44.4)	141 (35.5)
41+	52 (15.9)	45 (16.1)	46 (11.6)
**Gender (n,%)**			
Female	236 (72.2)	204 (73.1)	281 (70.8)
Male	91 (27.8)	75 (26.9)	116 (29.2)
**Ethnicity (n,%)**			
Chinese	128 (39.1)	110 (39.4)	175 (44.1)
Filipino	92 (28.1)	88 (31.5)	95 (23.9)
Others	107 (32.7)	81 (29.0)	127 (32.0)
**Marital status (n,%)**			
Single/Separated/Divorced/Widowed	180 (55.1)	142 (50.9)	263 (66.3)
Married	147 (45.0)	137 (49.1)	134 (33.8)
**Occupation (n,%)**			
Senior doctor	25 (7.7)	23 (8.2)	25 (6.3)
Junior doctor	64 (19.6)	38 (13.6)	70 (17.6)
Nurse	217 (66.4)	206 (73.8)	249 (62.7)
Healthcare Assistant	21 (6.4)	12 (4.3)	53 (13.4)
**Past medical history (n,%)**			
Yes	14 (4.3)	15 (5.4)	22 (5.5)
No	313 (95.7)	264 (94.6)	375 (94.5)
**Living with young children (<12 years) (n,%)**			
Yes	50 (15.3)	60 (21.5)	70 (17.6)
No	277 (84.7)	219 (78.5)	327 (82.4)
**Living with elderly (>65 years) (n,%)**			
Yes	51 (15.6)	45 (16.1)	65 (16.4)
No	276 (84.4)	234 (83.9)	332 (83.6)
**Lives alone (n,%)**			
Yes	44 (13.5)	47 (16.9)	49 (12.3)
No	283 (86.5)	232 (83.1)	348 (87.7)
**Practices a religion (n,%)**			
Yes	251 (76.8)	213 (76.3)	304 (76.6)
No	76 (23.3)	66 (23.7)	93 (23.4)
**Has family/close friend with COVID-19 (n,%)**			
Yes	25 (7.7)	48 (17.2)	330 (83.1)
No	302 (92.3)	231 (82.8)	67 (16.9)

**Table 2 ijerph-19-16878-t002:** The mean scores for the different categories of ED HCWs’ concerns and perceptions for the three cohorts. Comparison of mean scores of ED HCWs’ concerns and perceptions within the subgroups for the three cohorts.

Concerns and Perceptions		2020 (Mean ± SD)	2021 (Mean ± SD)		2022 (Mean ± SD)	
Concerns about infection risk		4.19 ± 0.82	3.93 ± 0.83		3.90 ± 0.92	
Concerns about working environment		4.09 ± 0.85	3.96 ± 0.98		3.90 ± 1.03	
Concerns about workload		4.04 ± 0.97	4.37 ± 0.89		4.81 ± 0.86	
Social connectedness		4.53 ± 0.64	4.43 ± 0.67		4.57 ± 0.70	
Workplace support		4.80 ± 0.69	4.66 ± 0.70		4.48 ± 0.76	
**Concerns & perceptions**	**Senior doctors**			**Junior doctors**		
	**2020** **(n = 25)**	**2021** **(n = 23)**	**2022** **(n = 25)**	**2020** **(n = 64)**	**2021** **(n = 38)**	**2022** **(n = 70)**
Concerns about infection	3.83 ± 0.70	3.61 ± 0.74	3.59 ± 1.10	3.94 ± 0.76	3.85 ± 0.85	3.86 ± 0.85
Concerns about working environment	4.37 ± 0.60	4.37 ± 0.81	4.12 ± 1.11	3.91 ± 0.70	4.10 ± 0.81	4.28 ± 0.87
Concerns about workload	3.25 ± 0.79	3.95 ± 1.00	4.76 ± 0.92	3.45 ± 0.88	4.29 ± 0.82	4.86 ± 0.87
Social connectedness	4.79 ± 0.78	4.43 ± 0.79	4.47 ± 0.71	4.71 ± 0.65	4.52 ± 0.66	4.62 ± 0.87
Workplace support	4.83 ± 0.62	4.43 ± 0.88	4.40 ± 0.98	4.81 ± 0.60	4.71 ± 0.61	4.36 ± 0.63
**Concerns & perceptions**	**Nurses**			**HCAs**		
	**2020** **(n = 217)**	**2021** **(n = 206)**	**2022** **(n = 248)**	**2020** **(n = 21)**	**2021** **(n = 12)**	**2022** **(n = 53)**
Concerns about infection	4.33 ± 0.80	3.99 ± 0.83	3.94 ± 0.92	3.93 ± 1.00	3.72 ± 0.70	3.87 ± 0.89
Concerns about working environment	4.11 ± 0.89	3.88 ± 1.03	3.79 ± 1.05	4.15 ± 0.99	4.02 ± 0.90	3.81 ± 0.99
Concerns about workload	4.34 ± 0.86	4.48 ± 0.87	4.92 ± 0.79	3.69 ± 0.99	3.69 ± 0.78	4.21 ± 0.89
Social connectedness	4.48 ± 0.60	4.39 ± 0.67	4.54 ± 0.67	4.12 ± 0.65	4.69 ± 0.52	4.67 ± 0.59
Workplace support	4.80 ± 0.69	4.66 ± 0.70	4.46 ± 0.75	4.69 ± 1.05	5.05 ± 0.50	4.80 ± 0.77

**Table 3 ijerph-19-16878-t003:** Distribution of the different severities of depression, anxiety, stress and PTSD of clinical concern amongst all ED HCWs in the three cohorts. Depression, anxiety, stress and PTSD scores amongst all the ED HCWs and its subgroups in the three cohorts.

	All HCWs			Senior Doctors			Junior Doctors			Nurses			HCAs		
MHOs	2020 n = 327	2021 n = 279	2022 n = 397	2020 n = 25	2021 n = 23	2022 n = 25	2020 n = 64	2021 n = 38	2022 n = 70	2020 n = 217	2021 n = 206	2022 n = 249	2020 n = 21	2021 n = 12	2022 n = 53
**Depression**															
No	237 (72.5)	196 (70.3)	269 (67.8)	18 (72.0)	16 (69.6)	16 (64.0)	52 (81.3)	28 (73.7)	50 (71.4)	157 (72.4)	144 (69.9)	166 (66.7)	10 (47.6)	8 (66.7)	37 (69.8)
Yes	90 (27.5)	83 (29.7)	128 (32.2)	7 (28.0)	7 (30.4)	9 (36.0)	12 (18.7)	10 (26.3)	20 (28.6)	60 (27.6)	62 (30.1)	83 (33.3)	11 (52.4)	4 (33.3)	16 (30.2)
*Mild*	*38 (11.6)*	*34 (12.2)*	*52 (13.1)*	*2 (8.0)*	*2 (8.7)*	*5 (20.0)*	*5 (7.8)*	*4 (10.5)*	*8 (11.4)*	*28 (12.9)*	*24 (11.7)*	*34 (13.7)*	*3 (14.3)*	*4 (33.3)*	*5 (9.4)*
*Moderate*	*36 (11)*	*34 (12.2)*	*50 (12.6)*	*2 (8.0)*	*2 (8.7)*	*3 (12.0)*	*4 (6.3)*	*4 (10.5)*	*5 (7.1)*	*24 (11.1)*	*28 (13.6)*	*34 (13.7)*	*6 (28.6)*	*0 (0)*	*8 (15.1)*
*Severe*	*10 (3.1)*	*4 (1.4)*	*11 (2.8)*	*2 (8.0)*	*1 (4.4)*	*1 (4.0)*	*1 (1.6)*	*0 (0)*	*5 (7.1)*	*5 (2.3)*	*3 (1.5)*	*4 (1.6)*	*2 (9.5)*	*0 (0)*	*1 (1.9)*
*Extremely severe*	*6 (1.8)*	*11 (3.9)*	*15 (3.8)*	*1 (4.0)*	*2 (8.7)*	*0 (0)*	*2 (3.1)*	*2 (5.3)*	*2 (2.9)*	*3 (1.4)*	*7 (3.4)*	*11 (4.4)*	*0 (0)*	*0 (0)*	*2 (3.8)*
**Anxiety**															
No	215 (65.8)	199 (71.3)	244 (61.5)	22 (88.0)	19 (82.6)	18 (72.0)	47 (73.4)	31 (81.6)	52 (74.3)	140 (64.5)	145 (70.4)	147 (59.0)	6 (28.6)	4 (33.3)	27 (50.9)
Yes	112 (34.2)	80 (28.7)	153 (38.5)	3 (12.0)	4 (17.4)	7 (28.0)	17 (26.6)	7 (18.4)	18 (25.7)	77 (35.5)	61 (29.6)	102 (41.0)	15 (71.4)	8 (66.7)	26 (49.1)
*Mild*	*33 (10.1)*	*17 (6.1)*	*33 (8.3)*	*1 (4.0)*	*2 (8.7)*	*1 (4.0)*	*9 (14.1)*	*2 (5.3)*	*4 (5.7)*	*20 (9.2)*	*13 (6.3)*	*20 (8.0)*	*3 (14.3)*	*0 (0)*	*8 (15.1)*
*Moderate*	*49 (15)*	*37 (13.3)*	*71 (17.9)*	*2 (8.0)*	*2 (8.7)*	*5 (20.0)*	*5 (7.8)*	*3 (7.9)*	*10 (14.3)*	*36 (16.6)*	*25 (12.1)*	*43 (17.3)*	*6 (28.6)*	*7 (58.3)*	*13 (24.5)*
*Severe*	*10 (3.1)*	*12 (4.3)*	*26 (6.6)*	*0 (0)*	*0 (0)*	*1 (4.0)*	*2 (3.1)*	*1 (2.6)*	*4 (5.7)*	*8 (3.7)*	*10 (4.9)*	*20 (8.0)*	*0 (0)*	*1 (8.3)*	*2 (3.8)*
*Extremely severe*	*20 (6.1)*	*14 (5)*	*23 (5.8)*	*0 (0)*	*0 (0)*	*0 (0)*	*1 (1.6)*	*1 (2.6)*	*0 (0)*	*13 (6.0)*	*13 (6.3)*	*19 (7.6)*	*6 (28.6)*	*0 (0)*	*3 (5.7)*
**Stress**															
No	287 (87.8)	240 (86.0)	328 (82.6)	22 (88.0)	20 (87.0)	23 (92.0)	61 (95.3)	33 (86.8)	59 (84.3)	191 (88.0)	176 (85.4)	200 (80.3)	13 (61.9)	11 (91.7)	46 (86.8)
Yes	40 (12.2)	39 (14.0)	69 (17.4)	3 (12.0)	3 (13.0)	2 (8.0)	3 (4.7)	5 (13.2)	11 (15.7)	26 (12.0)	30 (14.6)	49 (19.7)	8 (38.1)	1 (8.3)	7 (13.2)
*Mild*	*15 (4.6)*	*18 (6.5)*	*27 (6.8)*	*1 (4)*	*2 (8.7)*	*0 (0)*	*1 (1.6)*	*3 (7.9)*	*3 (4.3)*	*9 (4.2)*	*12 (5.8)*	*22 (8.8)*	*4 (19.1)*	*1 (8.3)*	*2 (3.8)*
*Moderate*	*16 (4.9)*	*15 (5.4)*	*25 (6.3)*	*2 (8)*	*1 (4.4)*	*1 (4.0)*	*1 (1.6)*	*1 (2.6)*	*5 (7.1)*	*11 (5.1)*	*13 (6.3)*	*17 (6.8)*	*2 (9.5)*	*0 (0)*	*2 (3.8)*
*Severe*	*8 (2.5)*	*3 (1.1)*	*12 (3)*	*0 (0)*	*0 (0)*	*1 (4.0)*	*0 (0)*	*0 (0)*	*3 (4.3)*	*6 (2.8)*	*3 (1.5)*	*7 (2.8)*	*2 (9.5)*	*0 (0)*	*1 (1.9)*
*Extremely severe*	*1 (0.3)*	*3 (1.1)*	*5 (1.3)*	*0 (0)*	*0 (0)*	*0 (0)*	*1 (1.6)*	*1 (2.6)*	*0 (0)*	*0 (0)*	*2 (1.0)*	*3 (1.2)*	*0 (0)*	*0 (0)*	*2 (3.8)*
**PTSD of Clinical Concern**															
No	274 (83.8)	241 (86.4)	333 (83.9)	22 (88)	21 (91.3)	23 (92.0)	59 (92.2)	35 (92.1)	63 (90.0)	181 (83.4)	176 (85.4)	203 (81.5)	12 (57.1)	9 (75.0)	43 (81.1)
Yes	53 (16.2)	38 (13.6)	64 (16.1)	3 (12)	2 (8.7)	2 (8.0)	5 (7.8)	3 (7.9)	7 (10.0)	36 (16.6)	30 (14.6)	46 (18.5)	9 (42.9)	3 (25.0)	10 (18.9)

**MHOs**		**2020** **Median (Q1-Q3)**		**2021** **Median (Q1-Q3)**			**2022** **Median (Q1-Q3)**		**2020** **Mean (±SD)**		**2021** **Mean (±SD)**			**2022** **Mean (±SD)**	
**All HCWs**		n = 327		n = 279			n = 397		n = 327		n = 279			n = 397	
Depression		2 (0–5)		2 (0–5)			2 (1–5)		3.3 ± 3.6		3.5 ± 4			3.7 ± 3.9	
Anxiety		2 (1–4)		2 (0–4)			2 (1–5)		3.1 ± 3.3		2.7 ± 3.3			3.4 ± 3.6	
Stress		3 (1–6)		3 (1–6)			4 (2–7)		3.8 ± 3.5		3.8 ± 3.6			4.6 ± 3.8	
PTSD		7 (2–18)		7 (2–16)			9 (2–20)		12.3 ± 14.5		11.1 ± 12.8			12.7 ± 12.8	
**Senior doctors**		n = 25		n = 23			n = 25		n = 25		n = 23			n = 25	
Depression		1 (0–5)		2 (0–6)			2 (1–5)		3.0 ± 4.4		4.2 ± 5.5			3.5 ± 3.4	
Anxiety		1 (0–2)		1 (0–3)			1 (0–4)		1.4 ± 1.8		1.4 ± 1.8			2.2 ± 3.0	
Stress		2 (1–5)		3 (1–6)			4 (1–6)		3.7 ± 3.4		3.5 ± 3.4			4.2 ± 3.5	
PTSD		6 (1–9)		3 (0–10)			3 (0–11)		8.7 ± 10.4		6.5 ± 8.4			7.5 ± 9.1	
**Junior doctors**		n = 64		n = 38			n = 70		n = 64		n = 38			n = 70	
Depression		1.5 (0–4)		3 (1–5)			2 (0–5)		2.7 ± 4.0		3.6 ± 4.0			3.3 ± 4.1	
Anxiety		2 (1–4)		1 (0–3)			2 (0–4)		2.5 ± 3.0		2.0 ± 2.9			2.4 ± 2.5	
Stress		3 (1–5.5)		4 (1–6)			3.5 (1–6)		3.5 ± 3.4		4.2 ± 3.6			4.3 ± 3.8	
PTSD		5 (1–9.5)		4 (1–10)			4.5 (0–16)		8.6 ± 13.7		9.0 ± 13.9			9.5 ± 11.7	
**Nurses**		n = 217		n = 206			n = 249		n = 217		n = 206			n = 249	
Depression		3 (1–5)		2 (0–5)			3 (1–6)		3.3 ± 3.3		3.5 ± 3.9			3.8 ± 3.9	
Anxiety		2 (1–5)		2 (0–4)			3 (1–6)		3.2 ± 3.2		2.9 ± 3.5			3.8 ± 3.9	
Stress		2 (1–6)		3 (1–5)			4 (2–7)		3.6 ± 3.4		3.7 ± 3.7			4.7 ± 3.8	
PTSD		7 (2–19)		8 (2–18)			10 (3–21)		12.5 ± 14.1		11.6 ± 13			13.6 ± 13.1	
**HCAs**		n = 21		n = 12			n = 53		n = 21		n = 12			n = 53	
Depression		5 (2–10)		2 (1–5)			2 (1–5)		5.8 ± 4.0		2.9 ± 2.0			3.6 ± 3.7	
Anxiety		6 (3–10)		5 (2.5–6)			3 (1–5)		6.0 ± 4.1		4.6 ± 2.3			3.9 ± 3.5	
Stress		7 (4–9)		4.5 (4–6)			3 (2–6)		6.7 ± 4.0		4.8 ± 1.9			4.4 ± 3.9	
PTSD		22 (13–45)		15 (11–23.5)			14 (6–20)		25.1 ± 17.5		17.3 ± 9.7			15.5 ± 13.1	

**Table 4 ijerph-19-16878-t004:** The association between individual factors and each MHO status using GEE.

	Depression		Anxiety		Stress		PTSD of Clinical Concern	
	OR (95% CI)	*p*-Value	OR (95% CI)	*p*-Value	OR (95% CI)	*p*-Value	OR (95% CI)	*p*-Value
**Survey Year** (Ref: 2020)								
2021	0.88 (0.62, 1.24)	0.461	0.76 (0.56, 1.02)	0.071	0.84 (0.55, 1.28)	0.412	0.70 (0.44, 1.11)	0.130
2022	0.62 (0.38, 1.01)	0.055	0.78 (0.49, 1.24)	0.289	0.75 (0.37, 1.52)	0.426	0.70 (0.35, 1.38)	0.301
**Gender** (Ref: Male)								
Female	1.04 (0.68, 1.59)	0.876	1.86 (1.20, 2.89)	0.005	1.30 (0.77, 2.20)	0.330	1.56 (0.84, 2.89)	0.162
**Age group** (Ref: 21–30)								
31–40	1.21 (0.79, 1.84)	0.382	0.66 (0.45, 0.97)	0.036	0.90 (0.54, 1.51)	0.692	0.68 (0.4, 1.15)	0.147
≥41	1.76 (0.84, 3.71)	0.135	0.93 (0.44, 1.95)	0.847	0.62 (0.25, 1.55)	0.306	1.00 (0.43, 2.29)	0.990
**Ethnicity** (Ref: Chinese)								
Filipino	0.72 (0.42, 1.21)	0.212	0.69 (0.41, 1.16)	0.165	0.51 (0.25, 1.02)	0.058	0.97 (0.49, 1.92)	0.925
Others	1.11 (0.72, 1.72)	0.641	1.24 (0.82, 1.87)	0.306	1.08 (0.64, 1.82)	0.779	1.69 (0.96, 2.96)	0.069
**Marital status** (Ref: Single/divorced/widowed)								
Married	0.70 (0.49, 1.02)	0.060	0.92 (0.64, 1.32)	0.648	1.16 (0.72, 1.87)	0.538	1.45 (0.91, 2.30)	0.117
**Occupation** (Ref: Senior doctors)								
Junior doctors	0.43 (0.19, 0.99)	0.048	0.67 (0.31, 1.46)	0.312	0.65 (0.2, 2.14)	0.475	0.68 (0.22, 2.10)	0.503
Nurses	0.63 (0.29, 1.38)	0.246	1.00 (0.48, 2.10)	0.999	0.86 (0.27, 2.75)	0.804	0.98 (0.34, 2.87)	0.976
HCAs	1.07 (0.42, 2.73)	0.886	2.29 (0.97, 5.40)	0.058	1.60 (0.43, 5.90)	0.482	2.33 (0.72, 7.60)	0.159
**Number of years in occupation**	0.94 (0.90, 0.98)	0.005	0.97 (0.93, 1.01)	0.100	0.95 (0.9, 1.01)	0.098	0.96 (0.91, 1.00)	0.06
**Psychiatric history**	3.75 (1.41, 9.96)	0.008	1.56 (0.48, 5.08)	0.460	3.20 (1.15, 8.92)	0.027	0.91 (0.24, 3.48)	0.895
**Living with elderly**	1.82 (1.20, 2.77)	0.005	1.44 (0.95, 2.19)	0.090	1.71 (1.08, 2.70)	0.022	2.12 (1.32, 3.40)	0.002
**Family infected by Covid**	1.28 (0.82, 1.98)	0.275	1.23 (0.80, 1.89)	0.349	1.01 (0.59, 1.73)	0.978	0.96 (0.53, 1.74)	0.897
**Workplace support**	0.74 (0.57, 0.96)	0.021	1.09 (0.85, 1.40)	0.496	1.13 (0.81, 1.57)	0.479	1.02 (0.74, 1.40)	0.925
**Social connected**	0.5 (0.38, 0.64)	<0.001	0.67 (0.52, 0.85)	0.001	0.57 (0.44, 0.75)	<0.001	0.52 (0.39, 0.7)	<0.001
**Concerns about infection**	1.15 (0.93, 1.41)	0.200	1.31 (1.06, 1.62)	0.011	0.91 (0.71, 1.17)	0.455	1.48 (1.15, 1.92)	0.003
**Concerns about working environment**	1.21 (1.03, 1.44)	0.024	1.28 (1.08, 1.51)	0.004	1.14 (0.93, 1.41)	0.204	1.16 (0.92, 1.47)	0.213
**Concerns about workload**	1.46 (1.19, 1.79)	<0.001	1.22 (1.00, 1.48)	0.050	1.9 (1.41, 2.55)	<0.001	1.64 (1.25, 2.15)	<0.001
**Agreed religion help cope with stress**	0.84 (0.53, 1.32)	0.448	1.18 (0.76, 1.82)	0.466	0.59 (0.34, 1.03)	0.063	0.70 (0.37, 1.30)	0.259
**Agreed exercise help cope with stress**	0.91 (0.57, 1.44)	0.678	0.99 (0.64, 1.54)	0.972	0.99 (0.52, 1.88)	0.965	0.65 (0.38, 1.13)	0.127
**Feel respected**	0.63 (0.42, 0.95)	0.026	0.56 (0.38, 0.84)	0.005	0.56 (0.34, 0.93)	0.026	0.69 (0.41, 1.18)	0.173

Note: Only individuals who completed at least two measurements in any two survey years were included in the model. HCA: Healthcare Assistants; PTSD: Post-traumatic stress disorder; OR: odds ratio; 95% CI: 95% confidence interval.

## Data Availability

The individual datasets collected and analysed will not be publicly made available due to privacy and confidentiality reasons. Data presented in this study is available upon request from the corresponding author.

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
