# Peer review of "Longitudinal Study Comparing Mental Health Outcomes in Frontline Emergency Department Healthcare Workers through the Different Waves of the COVID-19 Pandemic"

_ijerph, 2022, doi:10.3390/ijerph192416878_

Round 1
Reviewer 1 Report
This study pertains to a very important topic, that deserves to be published. However, the following comments should first be addressed:
1. Introduction:
This part of the paper presents no theoretical basis and refers to only a few empirical studies to justify the topic and choice of variables (depression, anxiety, stress, and PTSD) as manifestations of the mental health of Healthcare Workers from the Emergency Department. It would be beneficial to show a bit stronger theoretical or empirical support for the choice of such variables, based on the professional literature.
The text in line 60 refers to Table A1 (Appendix A). In Table A1 there are many abbreviations without any explanations, but in scientific papers, all abbreviations should be explained – so together with (or instead of) abbreviations the full names (labels) should be used (e.g. MOH, VTL, ARC).
In the paper 2 aims of the study were declared:
1) to quantify changes in MHOs (depression, anxiety, stress, and post-traumatic stress disorder (PTSD) symptoms) in our cohort of frontline ED HCWs over the past 3.5 years;
2) to identify the factors longitudinally associated with poorer MHOs.
However, there are 3 objectives, because the first aim is not clearly formulated and concerns two different issues that should be divided into 2 research goals: a) description and differences in the perception of concerns and mental health between the 3 cohorts of ED employees (working in 2020, 2021 or 2022 – now these data are presented in tables in the text of paper) and b) description and changes in the perception of concerns and mental condition of employees who were working in all 3 different phases of the pandemic and took part in 3 surveys (now these data from the “matched HCWs” are presented in the annexes – from Appendix E to Appendix I).
Therefore, the goal should be clearly reformulated to include only 2 goals (in which case the data from "matched HCW" should be removed), or 3 goals should be distinguished. In any case, the Introduction section should clearly explain why it is important to study selected mental health manifestations and perceived concerns.
2. Materials and Methods
Line 73: the text “including the number of P1 and P2 acuity patients per month” is incomprehensible - clarification is needed here on what Patients P1 and P2 mean.
2.3. Statistical Analysis
In the article, no statistical tests were used to quantify differences or changes.
Line 119-120: “The distribution of the severity and status of each MHO were visually compared based on scores and percentages without using statistical tests as the data were partially dependent.” This sentence pertains only to the 3 cohorts, but does not concern the data from the dependent group of “matched HCWs” – they need to use tests for the dependent group.
Moreover, descriptive data allow only to describe of concerns and MHOs of the 3 groups (3 cohorts) or concerns and MHOs of “matched HCWs” in 3 surveys and do not allow to conclude about significant differences between the 3 cohorts of HCWs or about changes (especially, if the groups are rather small or very small, e.g. n=15 or even n=7) in 3 measures among the “matched HCWs”.
These two goals on differences and changes require different statistical tests: tests on the 3 cohorts as rather independent groups to quantify the differences between them, and tests for repeated measures (the 3 measures) in the dependent 'HCW matched' group to detect changes in perceptions of concern and in MHOs.
Line 121-122: To identify the factors longitudinally associated with poorer MHOs the Generalized Estimating Equations (GEE) employing binominal family and logit link function were performed. This information is incomplete - it requires at least one reference to the literature where this type of analysis is described and its advantages are shown.
3. Results
Vaguely formulated objectives and unclear presentation of data make it difficult to understand the results.
All data relevant to the 3 objectives should be presented in the text of the article. The authors use unusual table numbering - e.g. there is Table 2.A, 2.B, and Table A1 and A2 which is confusing. I propose using the typical numbering of tables in the article, marking them with consecutive Arabic numerals.
3.1. Demographics of the Cohorts
All conclusions about demographic differences between cohorts are not justified since they are derived from raw scores or percentages - here the appropriate chi-square tests to compare numbers of people or tests (t or z) comparing more than two proportions should be used.
3.2. Concerns and Perceptions of the Cohorts
Conclusions about differences in concerns between cohorts (Table 2.A) are not justified, as they are formulated solely on the basis of descriptive data.
They need evidence from comparisons of mean scores between the 3 cohorts, just as subgroup comparisons have been made for the three cohorts (Table 2.B). However, these data are still insufficient because they do not tell which cohorts differ.
Moreover, we cannot talk about trends if the p-value is greater than 0.10 (there are no differences or changes).
The conclusions about changes in the perception of concerns among HCWs who participated in the three surveys (n=160, Table A4 - Appendix F) need to support them with the results of statistical tests for repeated measures.
3.3. Mental Health Outcomes
Also, conclusions about differences in each MHO between cohorts (Table 3A and Table 3B) and about changes in the 3 surveys among matched WHCs (Table A5a and A5b) on the basis of descriptive data are not justified.
Moreover, why is the article presenting detailed data on the distribution of severity of each MHO in subgroups in 3 cohorts (Table 3A) and in subgroups in 3 measurements (Table A5a), if these data are not interpreted (I guess they are ambiguous and difficult to be interpreted) and are not discussed in the article at all.
The data presented in Table 4 clearly show that the odds of each MHO (depression, anxiety, stress, PTSD) do not significantly depend on the year of the study (the chances of developing each MHO were lower than in 2020, but the differences were not significant). This means that despite the transition to the endemic phase of COVID-19, the mental health of Healthcare Workers from the Emergency Department was rather low and similar in 3 cohorts (in years: 2020, 2021, and 2022) with the only tendency (p=0.055) to improve in 2022 (compared 2020) regarding depression.
Table A6 needs to be supplemented with p-values.
Discussion
The data in Table 4 are inconsistent with the statement in the Discussion section that this “study found (1) worsening depression and stress in the overall cohort” (lines 224-225), but these controversial results are not discussed anywhere.
Currently, the discussion on MHOs concerns unfounded conclusions drawn solely on the basis of descriptive data.
Reviewer 2 Report
The paper is interesting and relevant, the value is mainly in the data collected in the longitudinal study that enable to significantly support conclusions.
Minor aspects should be improved before considering it for publication:
- Title: please refer to Covid-19 pandemic explicitly; the concept of post-pandemic phase should be carefully reconsidered since also WHO declared the end of covid-19 pandemic is in sight but no international statement has been made on its official conclusion
- Introduction/Discussion: lines 54-60 Are those measures affecting in some way the results? If not can you provide an explaination based on references or motivated assumptions
- Methodology: lines 102-113 : are those scales/questions based on recognised questionnaire/established scales like the psychiatric ones?
- Acknowledgements should be provided to disclose the fact that some tables/figures are taken and implemented from your two previously published studies
-Discussions: clarification should be provided in terms of modification of workload based not only on Covid-19 cases but on the reintroduction of other cases that during the first waves of pandemic decreased
- Discussion: Is there a role of the built environment in those study? It seems authors are not aware of the modification of spaces that happened during the three years of study as well as the impact of the physical space on mental health, work conditions, HCW stress, burnout as several EBD studies showed in different settings. Working Environment (i.e. Appendix D) is not independent from the physical environment.
- Housing conditions has also been shown as a very important issue in the mental health of people; in addition to living with children/elderly, the housing conditions are very important. Recent studies evaluated with similar mental health scales the role of housing built environment.
- The main limitation is represented by the exclusion of socio-economic factors in this study. Several conclusions may be impacted by those aspects
- Additional references are needed to position the findings against or in line with literature and existing body of knowledge on the topic of Mental Health of Healthcare Workers during Covid-19, with specific regards to EDs. This can also support the limitations of the study
- Some statements are generic and lack of evidence/support in literature , i.e. line 274
- Conclusions: future development strategies should be highlighted. Is the study going to continue also next year?
Reviewer 3 Report
This is a fine paper with all the right features; there are only a few minor problems described in the attached report, otherwise this article is worth publishing.
